# Gene Mapping, Genome-Wide Transcriptome Analysis, and WGCNA Reveals the Molecular Mechanism for Triggering Programmed Cell Death in Rice Mutant *pir1*

**DOI:** 10.3390/plants9111607

**Published:** 2020-11-19

**Authors:** Xinyu Chen, Qiong Mei, Weifang Liang, Jia Sun, Xuming Wang, Jie Zhou, Junmin Wang, Yuhang Zhou, Bingsong Zheng, Yong Yang, Jianping Chen

**Affiliations:** 1State Key Laboratory of Subtropical Silviculture, Zhejiang A & F University, Hangzhou 311300, China; chenxinyu8262@gmail.com (X.C.); zyh19990515zyh@gmail.com (Y.Z.); bszheng@zafu.edu.cn (B.Z.); 2State Key Laboratory for Managing Biotic and Chemical Treats to the Quality and Safety of Agro-Products, Key Laboratory of Biotechnology for Plant Protection, Ministry of Agriculture, and Rural Affairs, Zhejiang Provincial Key Laboratory of Biotechnology for Plant Protection, Institute of Virology and Biotechnology, Zhejiang Academy of Agricultural Science, Hangzhou 310021, China; meiq17@stu.syau.edu.cn (Q.M.); xmwang@zaas.ac.cn (X.W.); zhoujie@zaas.ac.cn (J.Z.); wangjunmin@zaas.ac.cn (J.W.); 3Plant Pathogens Laboratory, College of Plant Protection, Shenyang Agricultural University, Shenyang 210095, China; 4College of Plant Protection, Yunnan Agricultural University, Kunming 650000, China; liangwf0102@gmail.com; 5College of Plant Protection, Fujian A & F University, Fuzhou 350002, China; sunjia8690@gmail.com; 6State Key Laboratory for Managing Biotic and Chemical Treats to the Quality and Safety of Agro-Products, Key Laboratory of Biotechnology for Plant Protection, Ministry of Agriculture, and Rural Affairs, Zhejiang Provincial Key Laboratory of Biotechnology for Plant Protection, Institute of Plant Virology, Ningbo University, Ningbo 315211, China

**Keywords:** PCD, gene mapping, transcriptome, lignin, hormone, WGCNA

## Abstract

Programmed cell death (PCD) is involved in plant growth and development and in resistance to biotic and abiotic stress. To understand the molecular mechanism that triggers PCD, phenotypic and physiological analysis was conducted using the first three leaves of mutant rice PCD-induced-resistance 1(*pir1*) and its wild-type ZJ22. The 2nd and 3rd leaves of *pir1* had a lesion mimic phenotype, which was shown to be an expression of PCD induced by H_2_O_2_-accumulation. The *PIR1* gene was mapped in a 498 kb-interval between the molecular markers RM3321 and RM3616 on chromosome 5, and further analysis suggested that the PCD phenotype of *pir1* is controlled by a novel gene for rice PCD. By comparing the mutant with wild type rice, 1679, 6019, and 4500 differentially expressed genes (DEGs) were identified in the three leaf positions, respectively. KEGG analysis revealed that DEGs were most highly enriched in phenylpropanoid biosynthesis, alpha-linolenic acid metabolism, and brassinosteroid biosynthesis. In addition, conjoint analysis of transcriptome data by weighted gene co-expression network analysis (WGCNA) showed that the turquoise module of the 18 identified modules may be related to PCD. There are close interactions or indirect cross-regulations between the differential genes that are significantly enriched in the phenylpropanoid biosynthesis pathway and the hormone biosynthesis pathway in this module, which indicates that these genes may respond to and trigger PCD.

## 1. Introduction

Programmed cell death (PCD) is a ubiquitous process in the development of organisms, a genetic mechanism for killing cells that is both active and orderly [1]. As a defense against pathogen infection and abiotic stress, PCD can ensure the dynamic balance of multicellular organisms. Its main manifestations are the loss of intercellular connections, cytoplasmic shrinkage, membrane vesiculation, DNA fragmentation, nuclear lysis, and the formation of apoptotic bodies [2].

PCD is essential for plants to maintain their cell growth and development because PCD mechanisms are often used in the expression of tissue and organ functions, and for efficient nutrition and reproduction [3]. The brassinosteroid biosynthetic pathway is activated before tracheary element PCD, and the synthesized brassinosteroids induce PCD and the formation of secondary walls [4,5]. Senescence in plant leaves or petals involves much PCD, and the triggering of PCD can lead to premature senescence [6]. *MADS29* is a key regulator of PCD during rice seed development. Rice plants with reduced *OsMADs29* expression showed an abnormal persistence of nucellar tissue and the atrophy of seeds. Under normal conditions, nucellar PCD is necessary to facilitate the supply of nutrients to the young embryo and endosperm after fertilization [7]. The developing microspores are supported by the ephemeral tapetum layer, and as they develop, the tapetum synthesizes and secretes the proteins and lipids for the pollen coat into the anther locule, which eventually triggers PCD and releases the high molecular-weight components of the pollen wall [8]. In *Arabidopsis*, the peripherally located lateral root cap (LRC) cells first divide and then differentiate and elongate until the LRC reaches the transition zone at the end of the meristem when the cells undergo PCD to control the location of the root cap and the size of the root cap organ [9].

In plant-pathogen interactions, PCD occurs during both the plant hypersensitive response (HR) to avirulent pathogen infection and in the susceptible reaction to virulent pathogen attack [10]. Most incompatible plant-pathogen interactions cause a biphasic oxidative burst, during which infected cells and neighboring cells trigger PCD to limit pathogen spread [11,12]. Inactivation of all or part of the mildew resistance locus O protein triggers PCD, which in turn up-regulates the response of the seedlings for the onset of pathogen defense [13]. In *Arabidopsis*, the two signaling genes *EDS1* and *PAD4* mediate part of the resistance response, regulating a reactive oxygen intermediate/salicylic acid-dependent defense signal amplification circuit regulated by the lesion stimulating *LSD1* gene [14]. Two rice lesion mimic mutants, *spl17* and *Spl26*, showed enhanced resistance to multiple strains of both *Magnaporthe oryzae*, the rice blast fungal pathogen, and *Xanthomonas oryzae* pv. *oryzae*, the bacterial blight pathogen [15].

At present, the mechanism by which plants trigger and regulate PCD has not been fully elucidated. Most studies have suggested that some plant hormone and reactive oxygen species (ROS) are involved in the regulation of plant PCD. For example, the plant hormone gibberellin (GA) stimulates the secretion of hydrolase and leads to an increase in H_2_O_2_ production in the barley aleurone cell, triggering PCD [16]. Due to rising H_2_O_2_ levels and down-regulation of antioxidant transcripts, the oxidative nature of rice anthers under drought leads to PCD and pollen production [17]. In early responses to ROS generation induced by hyperosmotic stress (NaCl and sorbitol), *Nicotiana tabacum* cells trigger PCD [18]. High temperature suppresses the enzyme-dependent ROS scavenging and induces a rise in the H_2_O_2_ level, which leads to PCD in tobacco cells [19]. Furthermore, knockdown of *OsAGO2* led to the over-accumulation of ROS and abnormal anther development, causing premature initiation of tapetal PCD [20]. *AtBAG6*, containing a central BCL-2-associated athanogene (*BAG*) domain, is a stress-up-regulated *Arabidopsis* CaM-binding protein, and the overexpression of *AtBAG6* induced PCD through ROS accumulation [21]. *ERH1* enhances *RPW8* transcription through the salicylic acid signaling pathway and leads to increased ceramide levels and subsequent accumulation, which triggers PCD in *Arabidopsis* [22]. However, despite an increasing number of studies on PCD, a deep understanding of the regulatory mechanism for plant PCD is still lacking.

In previous work, we constructed a rice mutant library through ethyl methane sulfonate (EMS) mutagenesis to a japonica rice (*Oryza sativa*) cultivar ZJ22. From the mutant library, we characterized a spontaneous lesion mimic mutant with higher resistance to rice bacterial blight disease than its wild type parent. Preliminary analysis showed that the lesions mimic phenotype and high disease resistance is caused by PCD in the mutant leaves and the mutant was provisionally designated as *pir1* (PCD-induced-resistance 1). In this study, we carried out phenotypic characterization, gene Mapping, RNA-seq analysis, and co-expression network analysis in *pir1* mutant and its wild type rice to study the novel regulatory mechanism for rice PCD.

## 2. Results

### 2.1. Characterization of Phenotype and Physiological Analysis

First of all, *pir1* plants were shorter than its wild type ZJ22 whether at the five-leaf stage or the adult stage (Figure 1a and Figure 2a,c). Especially at the adult stage, the height of *pir1* was approximately half of that of ZJ22. Compared with those of ZJ22, the fresh weight and dry weight of *pir1* were significantly decreased (Figure 1b,c). Furthermore, the growth period of *pir1* was about 12 days longer than that of wild type ZJ22 (Figure 1d).

It is interesting that *pir1* plants exhibited notable leaf symptom of lesion mimic. *pir1* showed lesion mimic spots in large numbers on the 3rd leaf and smaller numbers on the 2nd leaf, but hardly any on the flag leaf (Figure 2a,b). At the adult stage, *pir1* plants were more significantly different to the wild type rice in their phenotype (Figure 2c,d). The leaves of *pir1* had more striking lesion spots than at the five-leaf stage. These spots were reddish brown and almost covered the 2nd and 3rd leaves, with some milder symptoms on the flag leaf (Figure 2d).

Trypan Blue (TB) assays confirmed that there was cell death in the lesions seen. Deep blue spots were present on the 2nd and especially the 3rd leaf of *pir1*, while the flag leaf of *pir1*and the sampled leaves of ZJ22 were almost uniformly light blue (Figure 3a). In microscopic section, the mesophyll cells of the 2nd and 3rd leaves of *pir1* showed obvious damage and were remarkably stained blue in contrast to those of ZJ22 (Figure 3b,c). These results demonstrate conclusively that cell death occurred in the 2nd and 3rd leaves of *pir1*.

PCD in plants is often accompanied by ROS accumulation in cells, which can be detected by Diaminobenzidine (DAB) staining. After DAB staining, the leaves of *pir1* had many reddish-brown spots coinciding with the lesions and the cell death detected by TB staining (Figure 4a). In microscopy, the mesophyll cells in the top 2nd and 3rd leaves of *pir1* were also stained bright reddish brown, in contrast to the cells of ZJ22 (Figure 4b,c). These results indicate that ROS accumulated much more in the 2nd and 3rd leaves of *pir1* than in leaves of ZJ22.

### 2.2. Mapping of the Gene Locus for the Rice Mutant

For the genetic analysis, the mutant rice *pir1* was crossed with the wild-type rice ZJ22 and the *indica* cultivar 9311. The F_1_ progenies from these crosses did not have any lesions on their leaves. The F_2_ population showed segregation of the wild-type and lesion mimic phenotypes in a ratio of about 3:1 (χ^2^ < χ^2^_0.05_ = 3.84, *P* > 0.05) in both populations (Appendix A). These results indicate that the lesion mimic phenotype of *pir1* is controlled by a single recessive gene. The *PIR1* gene was initially mapped in the interval between markers RM6972 and RM26 on chromosome 5 using an F_2_ population containing 109 individual plants generated from the cross of *pir1* and 9311 (Figure 5). With a larger F_2_ population containing 1006 individuals from the cross of *pir1* and 9311, the target region was further narrowed down to the interval between markers RM3321 and RM3616, and the marker RM3790 was highly linked to the *PIR1* gene (Figure 5). The target region for *PIR1* gene was about 498 kb and comprised 65 predicted gene loci (Appendix A) according to the rice genome annotation project website (https://www.ncbi.nlm.nih.gov/genome/gdv/). None of the 65 predicted genes was similar to these cloned genes for the rice mutant with PCD (Appendix A), which suggests that *PIR1* gene is a novel gene for rice PCD.

In order to find the PIR1 target gene from the mapping chromosome interval, we analyzed the differential expression of the 65 predicted genes and identified the mutation in genes sequence between *pir1* and its wild type rice. FDR ≤ 0.05 and |log2FC| > 1 were taken as thresholds to judge whether these 65 predicted genes-differently expressed. In all comparison groups, significantly different expression was found in 22 among 65 predicted genes (Appendix A). For example, genes, such as ncbi_4339407 (ALA-interacting subunit 1), ncbi_4339367 (putative dual specificity protein phosphatase DSP8), and ncbi_4339387, ncbi_4339404 (UDP-glycosyltransferase 88F3), were up-regulated, while ncbi_107278226 (Uncharacterized protein) and ncbi_107278650 (pseudo histidine-containing phosphotransfer protein 5-like) were down-regulated in the comparison group between ZJ22b and *pir1*b. However, none of the SNPs or Indels was found in the sequence of anyone in the 65 candidate genes (Appendix A).

### 2.3. Evaluation of RNA-Seq Reads and Mapping Results

Leaves from three different leaf positions were collected for transcriptome analysis: the mutant’s flag leaf was designated as *pir1*a, the 2nd leaf (from the top) as *pir1*b, and the 3rd leaf as *pir1*c, while the wild type flag leaf was ZJ22a, the 2nd leaf ZJ22b, and the 3rd leaf ZJ22c, and each line comprises three biological duplication. The cDNA libraries of leaves were sequenced with eighteen samples using the Illumina deep-sequencing technique. A total of 42.1 to 56.4 million raw reads were generated from each cDNA library. Clean reads were obtained to 99.77–99.80%, after excluding the low-quality reads (Appendix A).

Then we used HISAT2 software to align high-quality libraries to the rice reference genome. Approximately 97% of clean reads were mapped to the reference genome, among which nearly 95% were mapped to unique locations (Appendix A). Meanwhile, the abundance of mapped transcripts was measured in terms of FPKM (fragment per kilobase of transcript per million mapped reads), and a total of 30,783 gene loci were detected among the samples. These results show that the transcriptome sequencing quality was sufficient for further analysis.

### 2.4. Sample Correlations

Consistency between samples was examined by calculating Pearson’s correlation coefficient (R value) between each sample pair. As shown in Appendix A, the minimum R value for comparisons between the three biological replicates for each sample was 94.9%, most of which were between 97.5% and 99.7%. Principal component analysis (PCA) also showed a close correlation in expression between replicate samples (Appendix A), confirming the reproducibility of the data and validating the subsequent analysis.

### 2.5. Analysis of Differentially Expressed Genes (DEGs)

For all treatments, a stringent valve of FDR < 0.05 and absolute value of |log_2_FC| > 1 were used as threshold for categorizing the DEGs and 1679, 6019, 4500, 5824, 6000, 1124, 1653, 3239, 424 differential genes were identified between ZJ22a and *pir1*a, ZJ22b and *pir1*b, ZJ22c and *pir1*c, ZJ22a and ZJ22b, ZJ22a and ZJ22c, ZJ22b and ZJ22c, *pir1*a and *pir1*b, *pir1*a and *pir1*c, and *pir1*b and *pir1*c, respectively (Appendix A). The total number of significantly regulated genes differed between the different comparisons (Figure 6). The largest difference was found between ZJ22b and *pir1*b (6019 significantly up-/down-regulated genes), followed by *pir1*a and *pir1*c (6000 significantly up-/down-regulated genes), and then *pir1*a and *pir1*b (5824 significantly up-/down-regulated genes). In addition, Figure 6a shows that a total of 4500 DGEs were identified between ZJ22c and *pir1*c, among which 2802 genes were significantly up-regulated, and 1698 were down-regulated. In general, the number of up-regulated genes was higher than that of down-regulated ones, the only exception being between *pir1*b and *pir1*c (Figure 6a). Volcano plots show that the number of up-and down-regulated genes had a distinct distribution pattern between the several comparisons, indicating clear global gene expression patterns between different comparisons (Figure 6b). For example, the distribution pattern of up-regulated genes between ZJ22 and *pir1* was much higher than that between the other comparisons. The four comparison groups ZJ22b versus *pir1*b, ZJ22c versus *pir1*c, *pir1*a versus *pir1*b, and *pir1*a versus *pir1*c had the most DEGs, indicating that the greatest differences in expression occurred in comparisons between leaves with and without PCD, as well as between leaves with slight and serious PCD.

### 2.6. Gene Ontology (GO) Analysis of the DEGs

GO analysis was done to functionally categorize the DEGs into three groups, i.e., Biological process, Cellular component, and Molecular function. In the different comparisons, there were similar distribution patterns with regard to the types of enriched GO terms (Figure 7, Appendix A). For example, 34 significantly enriched GO terms were identified in the ZJ22a and *pir1*a comparison, with 18 of them in biological processes, nine in cellular components, and seven in molecular functions. Most DEGs in the biological process group were classified into “metabolic process” (GO: 0008152), “cellular process” (GO: 0009987), and “single-organism process” (GO: 0044699). Of the cellular component group, “cell” (GO: 0005623), “cell part” (GO: 0044464), “organelle” (GO: 0043226), “membrane” (GO: 0016020), and “membrane part” (GO: 0044425) were the most abundant GO terms. Under the molecular function group, “catalytic activity” (GO: 0003824) and “binding” (GO: 0005488) were the most abundant GO terms, followed by “transporter activity” (GO: 0005215) (Figure 7). In these enriched GO terms, the number of differentially up-regulated genes was far more than that of down-regulated genes. In the ZJ22b and *pir1*b comparison, DEGs were divided into 44 GO terms, of which 21 were in biological processes, 12 cellular components, and 11 molecular functions. The comparison of ZJ22c with *pir1*c showed similar distribution patterns in the number and types of enrichment GO terms as the comparison of ZJ22b with *pir1*b. In the ZJ22a and ZJ22b comparison, DEGs were divided into 36 GO terms, including 18 different biological processes, ten cellular components, and eight molecular functions. In the comparison of ZJ22a with ZJ22c, 19 GO terms were enriched in biological process, nine GO terms were enriched in cellular component, and nine GO terms were enriched in molecular function. Furthermore, there were only 24 significantly enriched GO terms identified in the comparison of ZJ22b with ZJ22c, with GO terms for 12 biological processes, eight cellular components and four molecular functions. In the *pir1*a and *pir1*b comparison, 20 GO terms of the biological process group, 12 GO terms of the cellular component group, and 11 GO terms of the molecular function group were significantly enriched. In the *pir1*a and *pir1*c comparison, there were 42 GO terms, comprising 19 biological processes, 12 cellular components, and 11 molecular functions. The classification of GO enrichment in the comparison between *pir1*b and *pir1*c was similar to the comparison with the above, but, in each GO term, the number of differentially down-regulated genes was far greater than that of up-regulated genes (Figure 7).

### 2.7. KEGG Pathway Analysis of the DEGs

KEGG enrichment analysis of DEGs was then performed to better understand the pathways in which the identified DEGs were involved (Appendix A). The obviously enriched KEGG pathways are shown in Figure 8. In the comparison of three different leaf positions between ZJ22 and *pir1*, DEGs were enriched into similar pathways. DEGs were most highly enriched in metabolic pathways (KO01100), biosynthesis of secondary metabolites (KO01110), phenylpropanoid biosynthesis (KO00940), plant-pathogen interaction (KO04626), plant hormone signal transduction (KO04075), diterpenoid biosynthesis (KO00904), alpha-linolenic acid metabolism (KO00592), and brassinosteroid biosynthesis (KO00905). With regard to the DEGs involved in the phenylpropanoid biosynthesis, several genes were detected which were involved in the biosynthesis of lignin (Appendix A). The expression levels of those DEGs, including *CAD* (cinnamyl-alcohol dehydrogenase), *CCR* (cinnamoyl-CoA reductase), *PAL* (phenylalanine ammonia-lyase), and *4CL* (4-coumarate—CoA ligase), were significantly up-regulated in *pir1*. Moreover, genes associated with hormone synthesis were also significantly up-regulated in their respective pathways (Appendix A), for example, genes related to ethylene biosynthesis: *SAMS* (S-adenosylmethionine synthase), *ACS* (1-aminocyclopropane-1-carboxylate synthase), *ACO* (1-aminocyclopropane-1-carboxylate oxidase), genes related to jasmonic acid biosynthesis: *PLA* (phospholipase A), *AOS* (allene oxide synthase), *LOX* (lipoxygenase), *ACX* (acyl-CoA oxidase), and genes related to brassinolide biosynthesis: *DET2* (steroid 5-alpha-reductase), *CYP92A6* (cytochrome P450), and genes related to salicylic acid biosynthesis: *PAL* (phenylalanine ammonia-lyase). Furthermore, there were few pathways found to be enriched in comparisons between different leaf positions of ZJ22, and the degree of enrichment was usually less than that found in the other comparisons.

### 2.8. Identification of Common DEGs in All Comparisons between Wild Type and Mutant Rice

We further determined that there were 928 DEGs common to all the comparisons between the corresponding leaves of ZJ22 and *pir1* (ZJ22a vs. *pir1*a, ZJ22b vs. *pir1*b, ZJ22c vs. *pir1*c) (Appendix A). GO enrichment of these DEGs showed that 16 terms for biological processes, 9 for cellular components, and 7 for molecular functions were present in all of them (Appendix A). These represented genes related to metabolic processes, cellular processes, catalytic activity, binding, cell, and cell parts. KEGG expression enrichment analysis shows that these common DEGs were most highly enriched in plant–pathogen interaction, plant hormone signal transduction, diterpenoid biosynthesis, alpha-linolenic acid metabolism, mitogen-activated protein kinase (MAPK) signaling pathway-plant, phenylpropanoid biosynthesis, metabolic pathways, and biosynthesis of secondary metabolites (Appendix A).

### 2.9. Identification of Conserved DEGs via Weighted Gene Co-Expression Network Analysis (WGCNA)

In order to obtain a comprehensive understanding of the gene regulatory network and identify specific genes that respond to PCD in *pir1*, we performed WGCNA. The filtered genes were divided into 18 modules according to the similarity of their expression patterns (Appendix A). Further, we associated the gene expression profiles in each module with all the samples to generate a heat map of the module-sample matrix (Figure 9a). As anticipated, we found that the gene expression profiles in the turquoise module were remarkably different between samples with and without PCD (Figure 9b). This suggests that the genes in this module may trigger and participate in the PCD process.

The turquoise module contained 4015 genes that were mainly related to genetic information processing, carbohydrate metabolism, amino acid metabolism, and energy metabolism (Appendix A). To further study the molecular mechanism of PCD, the unigenes of this module were analyzed by GO enrichment and KEGG pathway. In GO analysis, the unigenes were mainly enriched in “metabolic process” (GO: 0008152), “cellular process” (GO: 0009987), “single-organism process” (GO: 0044699), “cell” (GO: 0005623), “cell part” (GO: 0044464), “organelle” (GO: 0043226), “catalytic activity” (GO: 0003824), and “binding” (GO: 0005488) (Figure 10a). In KEGG pathway analysis, the turquoise module was mainly involved in carbohydrate metabolism (KO01200), lipid metabolism (KO00592), biosynthesis of other secondary metabolites (KOo00940), translation (KO03010), and signal transduction (KO04075) (Figure 10b).

Next, the differential genes significantly enriched in phenylpropanoid biosynthesis and hormone biosynthesis pathways in the turquoise module were used as hub genes to construct a gene regulatory network (Figure 11) in order to investigate the co-expression network in the regulation of PCD, in which each node represents a gene, and the interconnected lines between genes represent co-expression interrelations. We observed that these hub genes have a close interaction or an indirect cross-regulation with each other, which indicates that these key genes may play a vital role in regulating the PCD process.

### 2.10. Validation of RNA-Seq Data by qRT-PCR Analysis

The transcriptional level of the tested genes based on qRT-PCR correlated very well with their respective abundance estimated by RNA-Seq (Figure 12), indicating that the RNA-Seq data obtained in this study are reliable and can support the transcriptomic analysis presented above.

## 3. Discussion

### 3.1. PIR1 Gene Is a Novel Gene for Rice PCD

The rice *pir1* mutant described and studied here displays a lesion mimic phenotype in its leaves (Figure 2). This phenotype in other plants is known to be caused by PCD and such mutants provide important material for studying the mechanism of PCD. TB staining showed that there was obvious cell death in the lesions and surrounding areas in *pir1* leaves (Figure 3), and DAB staining showed the accumulation of ROS in these leaves (Figure 4). It is well-known that ROS is associated with PCD, and it can therefore be concluded that the symptoms on *pir1* leaves are indeed the result of PCD.

Genetic analysis indicates that the lesion mimic phenotype of pir1 is controlled by a single gene (Appendix A), and the target *PIR1* gene was mapped to the 498 kb region that comprises 65 predicted gene loci (Figure 5). We tried to identify the possible candidate *PIR1* gene through analyzing the sequence mutation and differential expression of the 65 predicted genes. However, there are no SNPs or Indels in the sequence of these genes (Appendix A), indicating that the *PIR1*-related mutation caused by EMS treatment is probably located in the promoter sequence or intergenic region. In addition, 22 among these 65 predicted genes were found to differently expressed (Appendix A), however, these DEGs identification cannot help us to discover the candidate *PIR1* gene. Target mutant genes on the upstream of some regulatory pathways often exhibit little changed expression, while its regulated downstream genes often express with significantly different manner due to the regulatory cascade amplification. Taken together, the mapping region here is too wide to help us to identify the target *PIR1* gene. We have constructed larger mapping population containing 1137 individual plants to narrow down the mapping region and to identify and characterize the target *PIR1* gene, and these works will be presented in a future report. However, the present result of gene mapping shows that none of the 65 predicted genes confers rice PCD (Appendix A), indicating that the *PIR1* gene is a novel gene regulating rice PCD and suggesting that there might be a new regulatory mechanism for rice PCD in the mutant *pir1*. Transcriptome analysis was, therefore, then used to understand the gene regulatory mechanism of PCD in *pir1*.

### 3.2. Co-Expression Network in Regulation of PCD

By comparing *pir1* with ZJ22, 1679, 6019, and 4500 DEGs were identified in the three leaf positions, respectively. KEGG analysis revealed that DEGs were most highly enriched in phenylpropanoid biosynthesis, alpha-linolenic acid metabolism and brassinosteroid biosynthesis. This pathway enriched in the comparisons between different leaf positions of *pir1* were similar to those enriched in the comparisons between ZJ22 and *pir1* (Figure 8). This suggests that these pathways may be the key to causing PCD in rice leaves.

#### 3.2.1. Involvement of Lignin Synthesis-Related Genes in Regulation of Rice PCD

Secondary metabolites (such as lignin) are closely related to the PCD process, and the key enzymes in lignin synthesis, such as 4CL, PAL, CAD, and CCR, are essential for lignin synthesis [23,24,25,26,27,28]. It has been reported that overexpression of *OsAAE3*, a gene encoding 4CL, leads to an increase in H_2_O_2_ content, which triggers PCD induced by ROS [29]. As the first rate-limiting enzyme in the phenylpropane biosynthetic pathway, PAL gives rise to the changes in the cell wall polymer content after increasing PAL accumulation, which leads to the enhancement of the cell wall and triggers PCD [30]. It is also known that PAL and CAD inhibitors can reduce the occurrence of PCD [31]. In this study, the enrichment analysis of the pathways of the DGEs showed that the phenylpropanoid biosynthesis pathway was significantly enriched (Figure 8 and Appendix A). Additionally, the analysis demonstrated that the genes related to lignin synthesis including *PAL*, *C4H*, *4CL*, *CCR*, *CAD*, and *C3H* were up-regulated in *pir1* compared to ZJ22 (Appendix A), and this was confirmed by qRT-PCR analysis (Figure 12). These results suggest that increased levels of these enzymes might cause a cascade of PCD in *pir1*. We speculate that the increase of lignin synthesis-related enzymes would cause the increase of lignin content, which would lead to the change of polymer content in cell wall or trigger the production of ROS, followed by the oxidative crosslinking reaction, and ultimately trigger PCD in *pir1*.

#### 3.2.2. Involvement of Plant Hormone Synthesis-Related Genes in Regulation of Rice PCD

Many different signal molecules, such as plant hormones including jasmonic acid (JA), brassinosteroid (BR), salicylic acid (SA), and ethylene (ET), are closely related to the PCD process [32,33,34,35]. Lipoxygenase (LOX) is an essential enzyme for JA synthesis, and activation of LOX expression can significantly induce the synthesis and accumulation of JA [36,37]. Allene oxide synthase (AOS) has the universal characteristics of the CytP450 family and is another key enzyme in the JA synthesis pathway [38]. Previous studies have shown that the AOS promoter is activated by a variety of signals including JA and SA, suggesting that the regulation of AOS expression may play a major role in controlling JA signaling [39]. RNA-seq analysis here showed that the expression levels for genes of *LOX*, *AOS*, *ACX*, *OPR*, etc., were notably increased in *pir1* compared to those in ZJ22 (Appendix A), and the alpha-linolenic acid metabolism pathway was also significantly enriched (Figure 8 and Appendix A). Studies have shown that the endogenous JA initially mediates the sharp production of ROS by reducing the activity of enzymatic antioxidants [40]. On the other hand, Sagi et al. and Liu et al. found that exogenous JA treatment can lead to a significant increase in plasma membrane (PM) NADPH(mitogen-activated protein kinase) oxidase activity and there is evidence that PM NADPH oxidase is involved in the production of H_2_O_2_ [41,42].Therefore, we speculate that the increase in the expression of key enzymes for JA synthesis causes the accumulation of JA in *pir1*, which activates the activity of PM NADPH oxidase, which in turn causes a surge of H_2_O_2_, and ultimately triggers PCD. In addition, JA has an activating effect on mitogen-activated protein kinase (MAPK) cascades [43,44], and the activation of the MAPK pathway in cells may undermine the redox balance, resulting in ROS production and ultimately leading to cell death [45]. This study found that most of the genes encoding MAPK, such as *MPK3*, *MPK5*, *MPK13*, etc., were up regulated in *pir1* (Appendix A). The changes of these kinases may induce some cascade reactions in the mutant, which may result in the activation of partial downstream reactions or imbalance in homeostasis. Consequently, it can be convinced that JA may act as a signal molecule in *pir1* to activate MAPK, thereby causing downstream reaction, which leads to the accumulation of ROS, and further promote the occurrence of PCD.

Transcriptome analysis showed that the key genes of BR synthesis, such as *Det2*, *BR6OX1*, *CYP92A6*, *CYP90D2*, etc., were also significantly up regulated in *pir1* compared to ZJ22 (Appendix A). Moreover, the enrichment analysis revealed that the brassinosteroid synthesis pathway was significantly enriched (Figure 8 and Appendix A). Previous studies have found that BR signaling induces *RBOH1* expression and ROS production, which in turn directly triggers PCD and the degradation of tapetal cells during pollen development [46]. Similarly, by applying exogenous BR, NADPH oxidase may be activated to generate ROS following perception of the BR signal [47]. Kwon et al. also found that BRs may act as a signal to trigger cell death. RabG3b is activated by responding to BRs to form an autophagy structure, which ultimately leads to the gradual degradation of cell contents and organelles [48]. From this, we suppose that the increase in expression of the key genes for BR synthesis may induce a surge in ROS that triggers PCD in *pir1*. However, how BR induces ROS and regulates the downstream cascade in the PCD pathway is still unknown.

Studies have shown that ET has a regulatory effect on PCD [49,50]. The application of exogenous ET greatly stimulated camptothecin-induced H_2_O_2_ production and subsequent PCD [51]. ET promotes cadmium-induced death of tomato suspension cells, accompanied by excessively high H_2_O_2_ content [52]. Moeder et al. found that ozone induced ET biosynthesis, which caused H_2_O_2_ production and cell death, and suggested that ET may play an indispensable role in regulating cell death by amplifying the second ROS explosion [53]. Further, high levels of ET activate the NADPH oxidase complex, which leads to the generation of ROS and subsequent downstream process of PCD [51]. In addition, Trobacher et al. have found that ET activates its signal transduction pathway through the ET receptor to induce PCD-related genes expression and trigger PCD [54]. ET is known to be produced by the conversion of methionine to S-adenosylmethionine and then to 1-aminocyclopropane-1-carboxylic acid (ACC) through ACC synthase (ACS) and ACC oxidase (ACO), respectively [55]. Our research found that the gene expression levels of SAMS, ACS, and ACO, the key enzymes of ET biosynthesis, increased significantly in *pir1* (Appendix A), which might cause the increase in ET content. Taken together, the results here indicate that ET may stimulate the expression of PCD-related genes and/or enhance ROS generation by activating NADPH oxidase, which ultimately triggers PCD in *pir1*. On the other hand, It has been reported that Ca^2+^ is the second messenger induced by ET perception, and ET increases the cytoplasmic Ca^2+^ concentration by activating PM calcium channels, which in turn leads to the production of ROS [56]. In our study, a total of 48 DEGs related to calcium signaling were found, including calcineurin B-like proteins (*CBLs*), calcium uniporter protein (*MCU*), calmodulins (*CAMs*), and calcium-dependent protein kinases (*CDPKs*) (Appendix A). The differential expression of these calcium-related proteins contributes to the transduction of calcium signals and subsequently triggers the downstream cascade in the Ca^2+^ regulatory pathway. It is reported that the expression of Ca^2+^/Mg^2+^-dependent nucleases causes DNA fragmentation in the nucleus and other forms of apoptosis [57]. Moreover, studies have shown that changes in intracellular Ca^2+^ homeostasis can lead to the release of cytochrome c, which triggers PCD [58,59]. So, ET signaling can activate downstream Ca^2+^ signal-related genes to increase cytoplasmic Ca^2+^, which leads to the expression of PCD-associated genes, thereby also promoting PCD in *pir1*.

The expression of several genes encoding PAL, a key promoter of SA synthesis, increased significantly in *pir1* (Appendix A), which might lead to the accumulation of SA [60]. Low levels of SA trigger plant resistance and defense response, but high levels of SA can directly cause plant PCD and also accept the signals from upstream second messengers to trigger PCD, such as Ca^2+^ concentration, ROS signals, and CAMP signals [61]. Previous studies have shown that the production of ROS is closely related to changes in SA content [62]. Increasing the endogenous concentration of SA caused the rapid accumulation of ROS and increased the intracellular ROS level, which increased the oxidative damage and caused cell death in the tissue [63]. SA is believed to stimulate H_2_O_2_ production in two ways: SA acts as a regulator that enhances a protein phosphorylation-mediated or MAP kinase-dependent signaling pathway, which in turn activates NADPH oxidase activity to trigger an oxidative burst [64,65]. In addition, SA can directly inhibit enzymes (such as catalase) involved in the decomposition of H_2_O_2_, thereby increasing the level of H_2_O_2_ [66]. Transcriptome results showed that the genes encoding NADPH oxidase, such as *RBOHC*, *RBOHE*, *RBOHF*, etc., were significantly up-regulated in *pir1*, and it was found that more enzymes involved in the decomposition of H_2_O_2_, such as POD, CAT1, etc., were down-regulated in *pir1* (Appendix A). Furthermore, researchers have proposed that SA and H_2_O_2_ form a self-amplifying feedback loop. The initial increase of H_2_O_2_ activates the synthesis of SA, and then the subsequent increase in SA and ROS produces a synergistic effect to enhance cell death. At the same time, SA continues to enhance the production of H_2_O_2_, which in turn activates more SA synthesis and cell death [67]. It is reasonable to infer that the increase of SA content enhances H_2_O_2_ accumulation by activating the activity of NADPH oxidase or directly inhibiting the enzymes related to the decomposition of H_2_O_2_. In addition, due to the synergistic regulatory effect of these two signaling molecules, their content gradually accumulates, which then leads to oxidative damage and triggers cell death. Thereout, we speculate that SA accumulation is also an important trigger of PCD in *pir1*.

### 3.3. Working Model for PCD Regulation in pir1 Mutant

Based on these results, a putative working model for the triggering of PCD in *pir1* is proposed (Figure 13). In this model, the up regulation of genes related to lignin synthesis causes the accumulation of lignin that then triggers PCD. Additionally, several plant hormones may play an important role in promoting PCD in the rice mutant. Firstly, JA triggers PCD with ROS accumulation by inhibiting the activity of enzymatic antioxidants, such as catalase, and stimulating the activity of the PM NADPH oxidase or MAPK cascade. Secondly, genes related to BR synthesis are highly expressed thus increasing BR content. BR accumulation promotes ROS generation by activating some BR-participating signal pathways and ultimately triggers PCD in the rice mutant. Thirdly, the increase in ET content triggers PCD by activating the NADPH oxidase and subsequent ROS accumulation. Moreover, ET induces the expression of the PCD-related genes directly or by its role in the Ca^2+^ signal pathway to trigger PCD. Finally, SA triggers its signal transduction to suppress the activity of enzymatic antioxidants, such as catalase, and activating the activity of NADPH oxidase. Both of the pathways induce ROS accumulation and trigger PCD. Taken together, the results in the present work demonstrate that lignin and plant hormones (including JA, BR, ET, and SA) are mainly involved in triggering and regulating PCD in the rice mutant *pir1*.

## 4. Materials and Methods

### 4.1. Plant Materials

The wild type, *Oryza sativa* ssp. *Japonica* cultivar ZJ22, is maintained in our laboratory, and *pir1* is a lesion mimic mutant from an EMS mutant library that is derived from the ZJ22. Rice plants were sown in the seedling nursery and 25-day-old seedlings were transplanted into the experimental field in Zhejiang Academy of Agricultural Sciences, Hangzhou, China. In each experiment the plots were arranged in a complete randomized block design, with spaces of 20 cm between plants within each row and 35 cm between rows. Otherwise, field management followed normal agricultural practice. The leaves at three different positions, including the flag leaf, the 2nd leaf, and the 3rd leaf, of ZJ22 and *pir1* were collected for transcriptome analysis. Three biological replicates were set for each sample, and the leaves of three individual plants were taken as a replicate, and 25 mg leaves were selected from each individual plant at the five-leaf stage. At the five-leaf stage and the adult stage, 15 plants of each line were randomly chosen to detect agronomic traits. Plant growth period was calculated from the second day of seedlings raising to the mature stage.

### 4.2. Leaf Histochemical and Physiological Analyses

To detect and examine cell death, sampled leaves were cut into strips about 5 cm long, immersed in TB dye solution, placed on an electric stove covered with asbestos net, and heated and boiled for 10 min. After the dark blue spots had appeared, the tubes were cooled to room temperature and the leaf pieces were removed and then rinsed several times in hydrated trichloroacetaldehyde, followed by decolorization at room temperature for 24–48 h [68]. After the decolorization was completed, the blue spots (staining of dead cells) could be observed and assessed. For microscopic examination, decolorized leaf pieces were rinsed in 90% ethanol several times and then sectioned on a freezing microtome.

To assess ROS production, fresh leaves were immersed in a 1mg/L DAB solution. The solution was illuminated at 37 °C for 10 h, and then the leaves were boiled and decolorized in 90% ethanol until the chlorophyll had faded. After rinsing in fresh 90% ethanol for 2 h [69], frozen sections were taken and observed under an optical microscope.

### 4.3. Mapping of the Gene Locus for the Rice Mutant

Two F_2_ populations were developed by self-pollinating the F_1_ cross between the mutant rice *pir1* and the wild type rice ZJ22 and also the cross of *pir1* with an *indica* rice variety 9311. Genetic analysis was performed using 131 individual plants from the F_2_ population of *pir1* × ZJ22 and 109 from the F_2_ population of *pir1* × 9311. F_2_ plants from the *pir1* × 9311 cross were used for DNA marker and phenotype segregation analysis to map the gene locus. All the materials were sown in the seedling nursery and about 30-day-old seedlings were transplanted into the experimental field in Zhejiang Academy of Agricultural Science, Hangzhou, China. A total of 736 molecular markers, including simple sequence repeat (SSR, http://www.gramene.org/microsat/) and sequence-tagged site (STS, http://www.ncbi.nlm.nih.gov/) markers, were screened for polymorphism between the parents *pir1* and 9311, and 236 markers evenly distributed on 12 rice chromosomes were selected due to their polymorphism (Appendix A, Appendix A). The genetic linkage map of these polymorphic markers is shown in Appendix A. The initial mapping was conducted using these selected markers and an F_2_ population with 109 individuals. A larger F_2_ population with 1006 individuals and newly-developed markers were used to further narrow down the mapping interval on the chromosome.

### 4.4. Sampling and RNA Extraction

At the five-leaf stage, three independent duplicate samples were taken from each of the flag leaf, the 2nd leaf (from the top) and the 3rd leaf of both the mutant *pir1* and the wild type ZJ22. Total RNA was extracted using Trizol reagent kit (Invitrogen, Carlsbad, CA, USA) according to the manufacturer’s instructions. RNA quality was assessed on an Agilent 2100 Bioanalyzer (Agilent Technologies, Palo Alto, CA, USA) and checked using RNase-free agarose gel electrophoresis.

### 4.5. cDNA Library Construction and Sequencing

After extraction of total RNA, eukaryotic mRNA was enriched by Oligo (dT) beads, while prokaryotic mRNA was enriched by removing rRNA using the Ribo-ZeroTM Magnetic Kit (Epicentre, Madison, WI, USA). Then the enriched mRNA was fragmented into short fragments using fragmentation buffer and reverse transcribed into cDNA with random primers. Second-strand cDNAs were synthesized by DNA polymerase I, RNase H, dNTP, and buffer. Afterwards, we used QiaQuick PCR extraction kit (Qiagen, Venlo, The Netherlands) to purify the cDNA fragments and then performed end repaired, added poly (A), and ligated to Illumina sequencing adapters. The ligation products were size selected by agarose gel electrophoresis, amplified by PCR, and sequenced using Illumina HiSeq2500 by Gene Denovo Biotechnology Co (Guangzhou, China). The raw transcriptome data have been assigned the accession number PRJNA655984 in the NCBI sequence read archives (SRA) (https://www.ncbi.nlm.nih.gov/sra).

### 4.6. RNA-Seq Data Analysis

To obtain high quality clean reads, the raw sequencing reads were further filtered by fastp (version 0.18.0) [70]. The parameters were as follows: (1) removing reads containing adapters; (2) removing reads containing more than 10% of unknown nucleotides (N); (3) removing low quality reads containing more than 50% of low quality (Q-value ≤ 20) bases; and (4) removing sequences containing rRNA. The clean reads were aligned to the rice reference genome using HISAT2.2.4 with “-rna-strandness RF” and other parameters set as a default [71].

For each transcription region, a FPKM value was calculated to quantify its expression abundance and variations which then indicate the difference in expression between the two comparisons, using StringTie software [72,73].

Differentially expressed gene (DEG) analysis was performed by DESeq2 [74] software between two different comparison groups. The genes with the parameter of false discovery rate (FDR) below 0.05 and absolute fold change ≥2 (log_2_fold change ≥ 1 or log_2_fold change ≤ −1) were taken as thresholds to be considered differentially expressed.

GO [75] is an international standardized gene functional classification system which has three ontologies: molecular function, cellular component, and biological process. GO enrichment analysis gives the GO function classification annotation of the gene and screens for all the GO terms that are significantly enriched in DEGs compared to their genomic background, as well as filters the DEGs that correspond to specific biological functions. Firstly, all DEGs were mapped to GO terms in the Gene Ontology database (http://www.geneontology.org/), the number of genes in each term was calculated, and significantly enriched GO terms in DEGs compared to the genomic background were defined by a hypergeometric test. The calculated *p*-value was done with FDR ≤ 0.05 as a threshold through FDR Correction. Finally, it is concluded that GO terms meeting this condition were defined as significantly enriched compared with the entire genome background.

Genes usually interact with each other to play roles in certain biological functions. Meanwhile, pathway-based analysis helps to further understand the biological functions of genes. KEGG [76] is the major public pathway-related database [77]. Pathway enrichment analysis identifies significantly enriched metabolic pathways or signal transduction pathways in DEGs compared with the whole genome background, and then understand what biological functions are played. The calculated p-value was done through FDR Correction, taking FDR ≤ 0.05 as a threshold. Pathways meeting this condition were defined as significantly enriched.

### 4.7. Co-Expression Network Analysis for Module Construction

Co-expression networks were constructed using the WGCNA (v1.47) package in R [78]. After filtering approximately 45% of the genes, gene expression values were imported into WGCNA to construct co-expression modules with default settings except that the power was 8, TOM Type was unsigned, merge Cut Height was 0.85, and min Module size was 50. Genes were clustered into 18 correlated modules. To identify the relationship between module and Specific leaf position expression in the wild type and mutant, a correlation coefficient was calculated as a module eigenvalue with samples. Intramodular connectivity (K.in) and module correlation degree (MM) of each gene was calculated by the R package and genes with high connectivity tended to be hub genes which may have important functions. The networks were visualized using Cytoscape (v3.7.0) [79].

### 4.8. Validation of Gene Expression by qRT-PCR

qRT-PCR was used to validate the RNA-seq results using the Light Cycler^®^ 480 System (Roche, Basel, Kanton Basel, Switzerland) with 10 Figure final volumes containing 0.2 uL of cDNA, 0.2 μL of each Primer Premier, 5 μL of 2 × U1tra SYBR Mixture, and 4.4 μL RNase-free water, with the following amplification protocol: 94 °C for 5 min, followed by cycling for 30 rounds at 94 °C for 10 s, 60 °C for 10 s, and 72 °C for 20 s. The experiment was repeated three times using three biological replicates.

Eight genes that are important for lignin biosynthesis, sixteen genes involved in hormone biosynthesis, and four randomly chosen genes were selected (Appendix A). Actin was selected as the internal reference gene. The relative expression levels of the selected DEGs normalized to the expression level of the internal reference control were calculated using the 2^−ΔΔCt^ method [80]. The primers were designed with Primer Premier Software [81] and are listed in Appendix A.

## 5. Conclusions

Our results show that the lesion mimic phenotype in rice mutant *pir1* was caused by PCD, induced by H_2_O_2_ accumulation. The *PIR1* gene was mapped in a 498 kb interval between the molecular markers RM3321 and RM3616 on chromosome 5, and the PCD phenotype is controlled by a new gene conferring rice PCD. RNA-Seq analysis revealed that the phenylpropanoid pathway-related genes were up-regulated in *pir1* compared to the wild-type ZJ22, which may increase the biosynthesis of lignin to trigger the PCD. In addition, many genes related to the synthesis of plant hormones JA, BR, ET, and SA were differentially expressed in the leaves of *pir1*, suggesting that these plant hormones may be involved in triggering and regulating PCD in the rice mutant.

## Figures and Tables

**Figure 1 plants-09-01607-f001:**
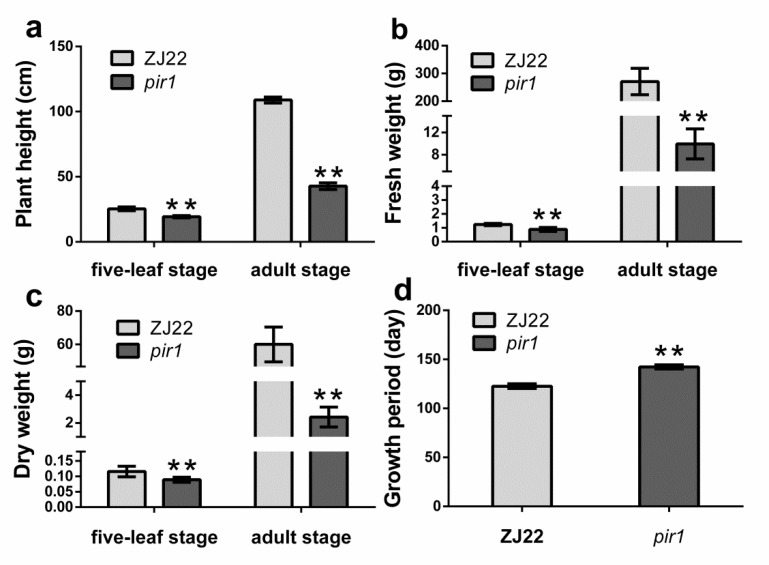
Comparison of agronomic traits between wild type ZJ22 and mutant PCD-induced-resistance 1 (*pir1*). (**a**) Plant height; (**b**) fresh weight; (**c**) dry weight; (**d**) growth period. Values are means ± SD of 15 plants, ** indicated significant differences at *p* < 0.01 determined by t-tests.

**Figure 2 plants-09-01607-f002:**
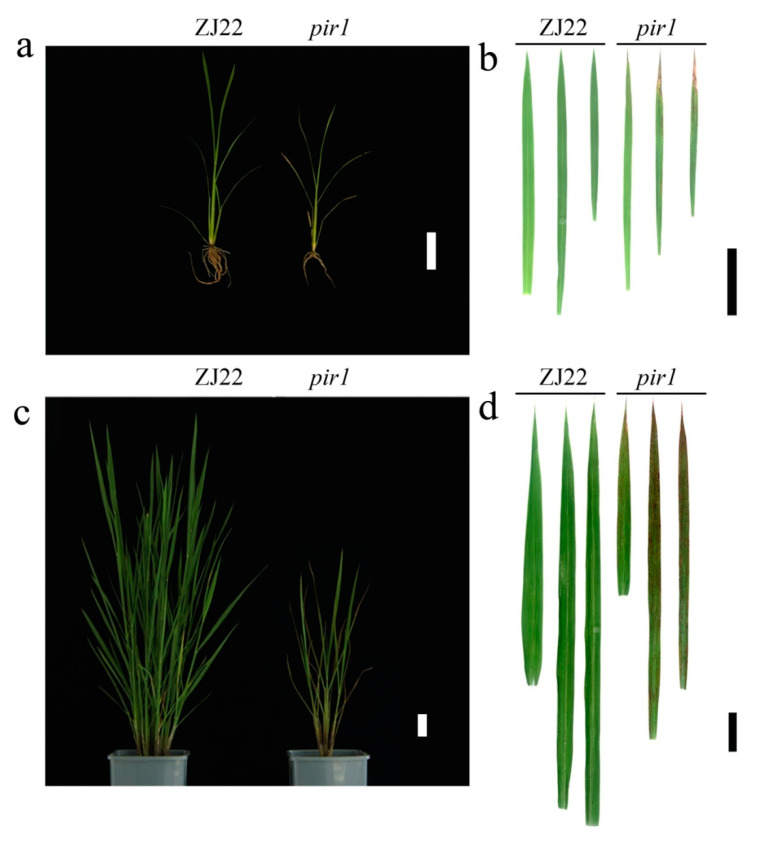
Comparison of the phenotypes of wild type ZJ22 and mutant *pir1*. (**a**,**c**) Whole plants at five-leaf (**a**) or adult (**c**) stages: left, ZJ22, and right, *pir1*. White scale bar = 5 cm. (**b**,**d**) Leaf phenotype at five-leaf (**b**) or adult (**d**) stages: the three leaves on the left represent the flag leaf, the 2nd leaf (from the top) and the 3rd leaf of ZJ22; the three leaves on the right represent the flag leaf, the 2nd leaf, and the 3rd leaf of *pir1*. Black scale bar = 2 cm.

**Figure 3 plants-09-01607-f003:**
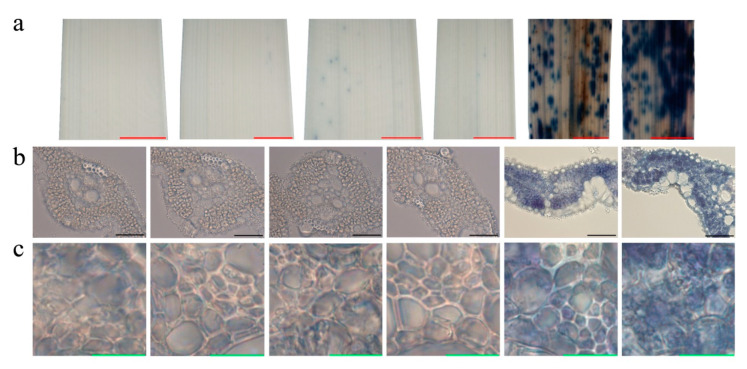
Trypan Blue (TB) staining on the leaves of ZJ22 and *pir1* sampled at the five-leaf stage. Each set of images represents (from left to right), the flag, 2nd and 3rd leaves of ZJ22, and then the flag, 2nd and 3rd leaves of *pir1*. (**a**) Leaf phenotype; red scale bar = 0.2 cm. (**b**) Cell morphology; black scale bar = 50 μm. (**c**) Magnified cell morphology; green scale bar = 10 μm.

**Figure 4 plants-09-01607-f004:**
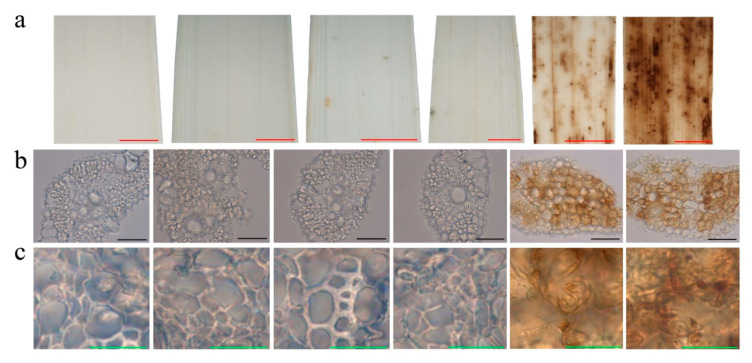
Diaminobenzidine (DAB) staining on the leaves of ZJ22 and *pir1* sampled at the five-leaf stage. Each set of images represents (from left to right), the flag, 2nd and 3rd leaves of ZJ22, and then the flag, 2nd and 3rd leaves of *pir1*. (**a**) Leaf phenotype; red scale bar = 0.2 cm. (**b**) Cell morphology; black scale bar = 50 μm. (**c**) Magnified cell morphology; green scale bar = 10 μm.

**Figure 5 plants-09-01607-f005:**
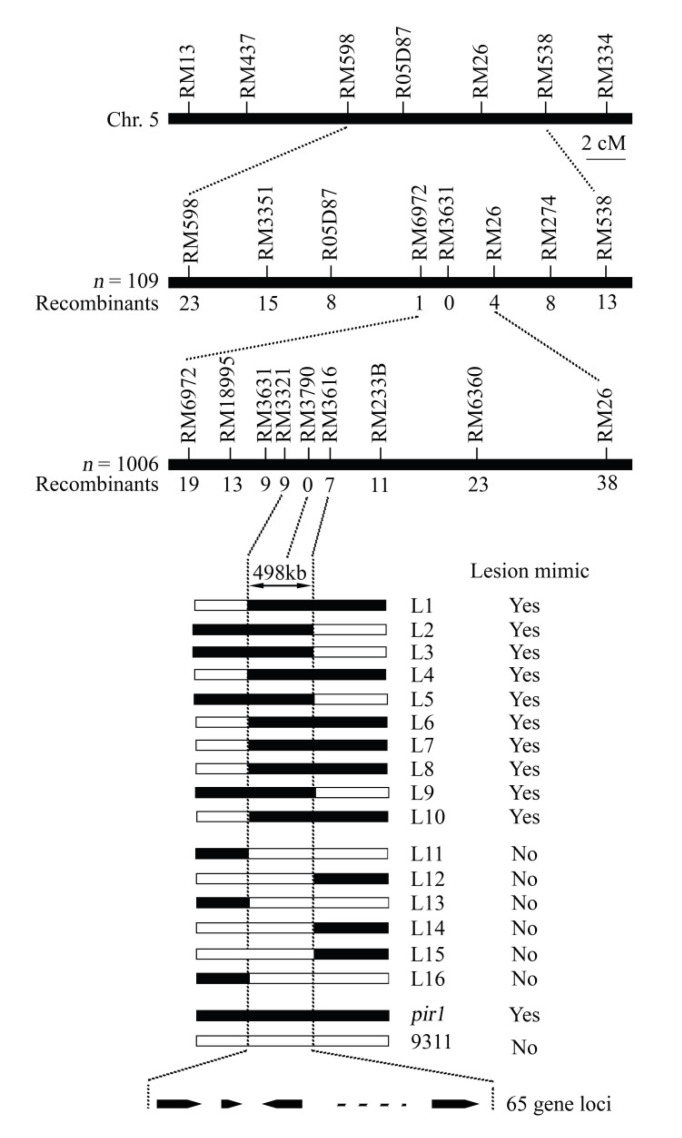
Mapping of the *PIR1* gene. *PIR1* locus was firstly mapped in the interval between markers RM6972 and RM26 on chromosome 5 and was further delimited to the interval between markers RM3321 and RM3616 by enlarging the mapping population from one of 109 individuals to 1006 individuals. The number of recombinants is indicated.

**Figure 6 plants-09-01607-f006:**
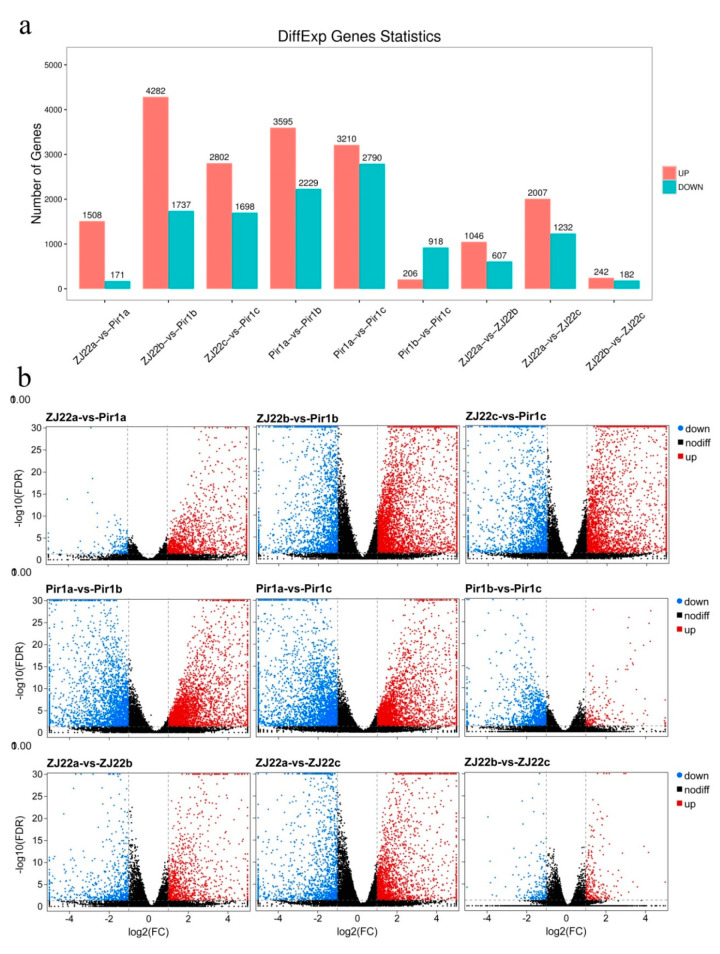
Analysis of differential gene expression in all pairwise comparisons. (**a**) The number of up-and down-regulated differentially expressed genes (DEGs). (**b**) Gene expression levels in volcano plots.

**Figure 7 plants-09-01607-f007:**
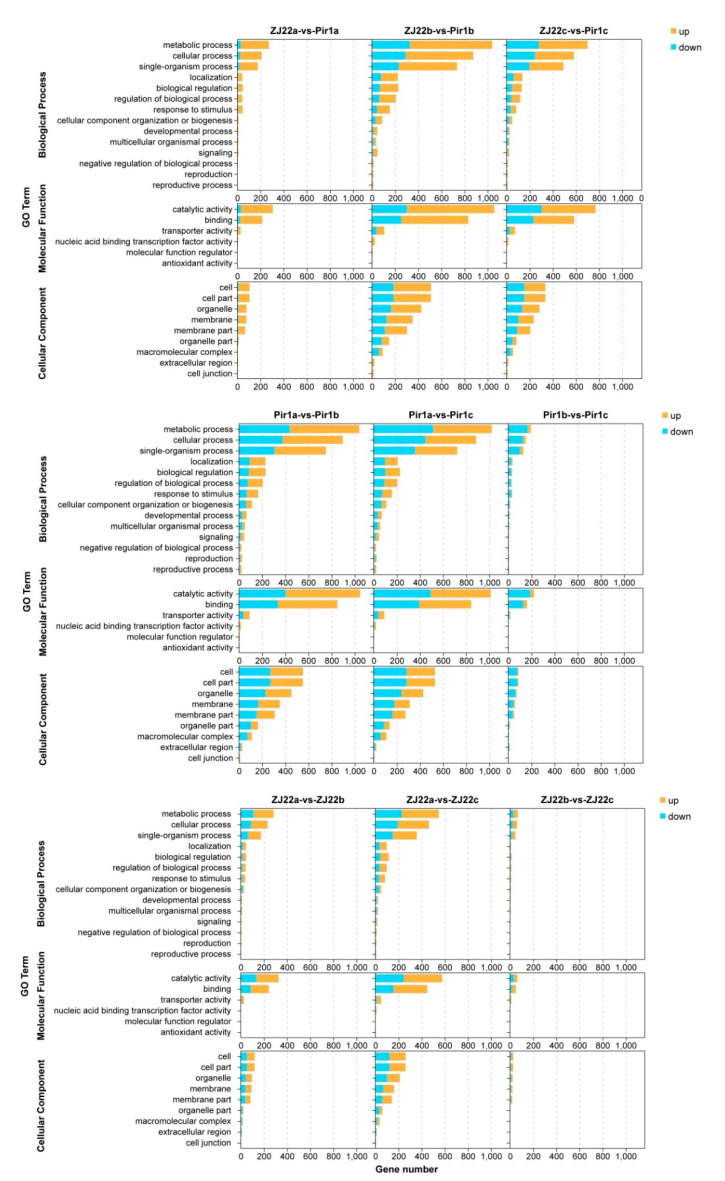
Summary of the distribution and number of DEGs in the three ontology classes, molecular function, cellular component, and biological process.

**Figure 8 plants-09-01607-f008:**
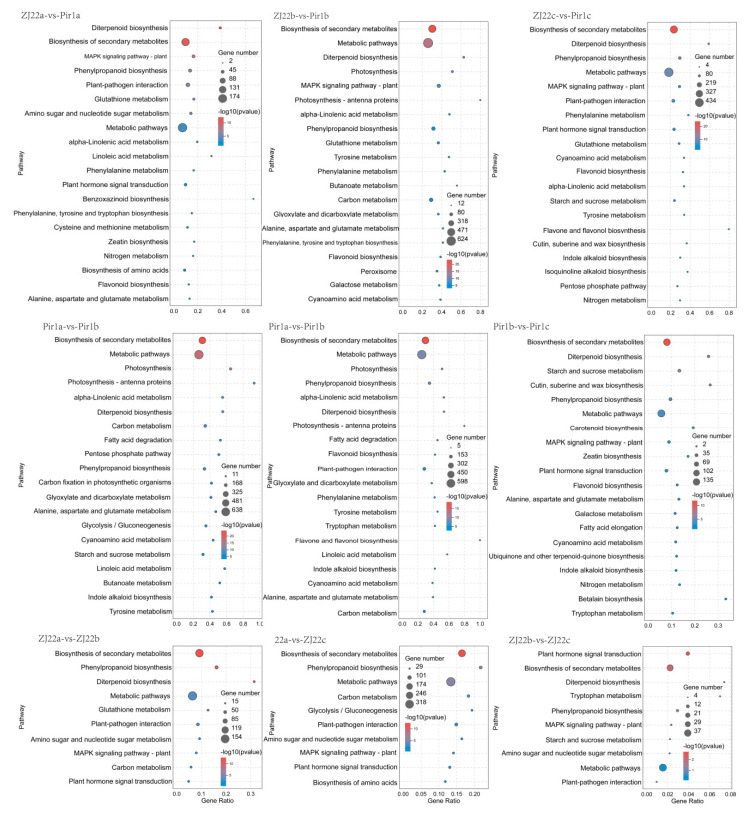
KEGG enrichments of the annotated DEGs. The left *Y*-axis indicates the KEGG pathway, and the *X*-axis indicates the rich factor. A high q-value is represented by blue, and a low q-value by red color. The rich factor is the ratio of the number of DEGs mapped to a certain pathway to the total number of genes mapped to this pathway.

**Figure 9 plants-09-01607-f009:**
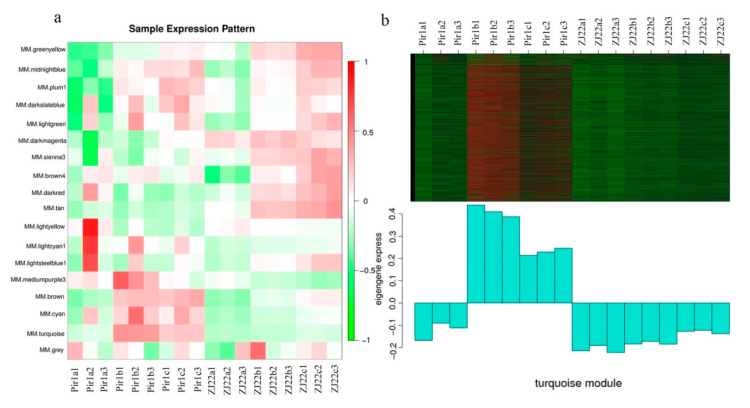
Sample expression pattern and Expression profile. (**a**) Heat map showing the sample expression patterns in the modules. (**b**) The expression profile of all the co-expressed genes in module turquoise. The color scale represents the *Z*-score. The bar graph shows the consensus expression pattern of the corresponding co-expressed genes in this module.

**Figure 10 plants-09-01607-f010:**
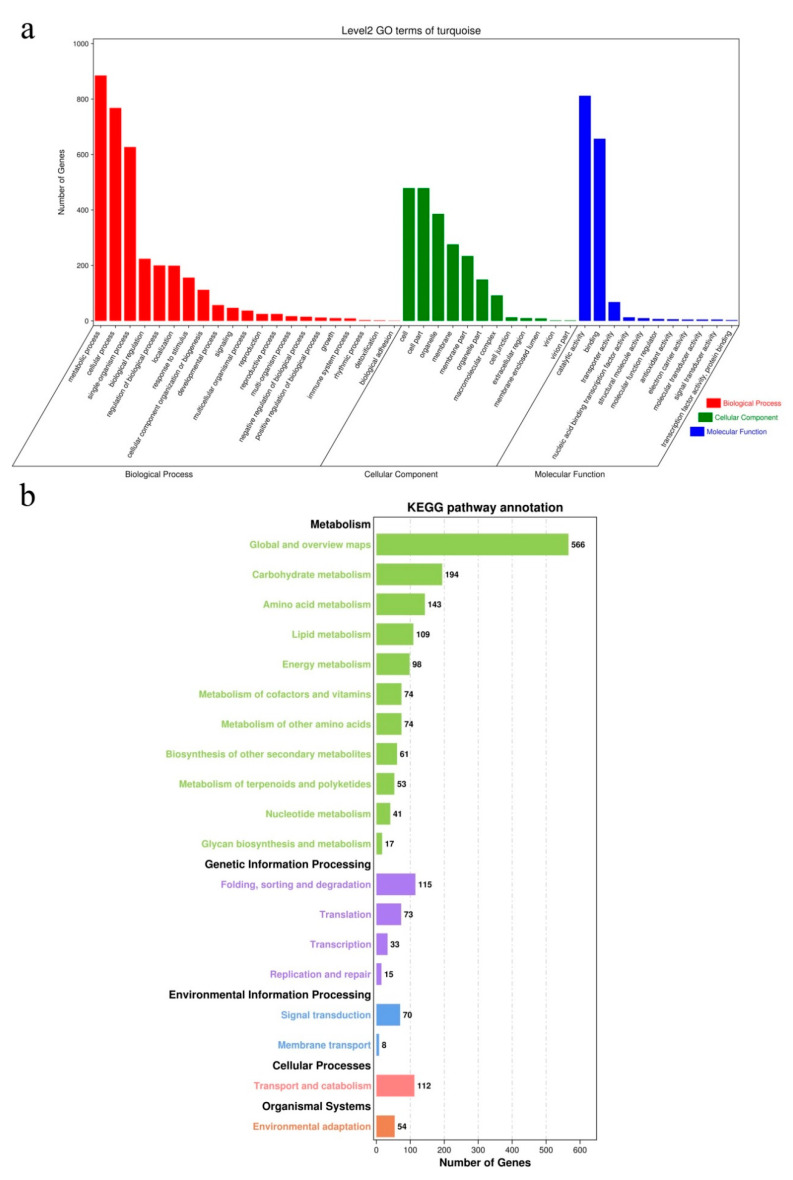
Function enrichment analysis of module turquoise gene. (**a**) Gene Ontology (GO) enrichment of module turquoise gene. (**b**) KEGG pathway enrichment of module turquoise gene.

**Figure 11 plants-09-01607-f011:**
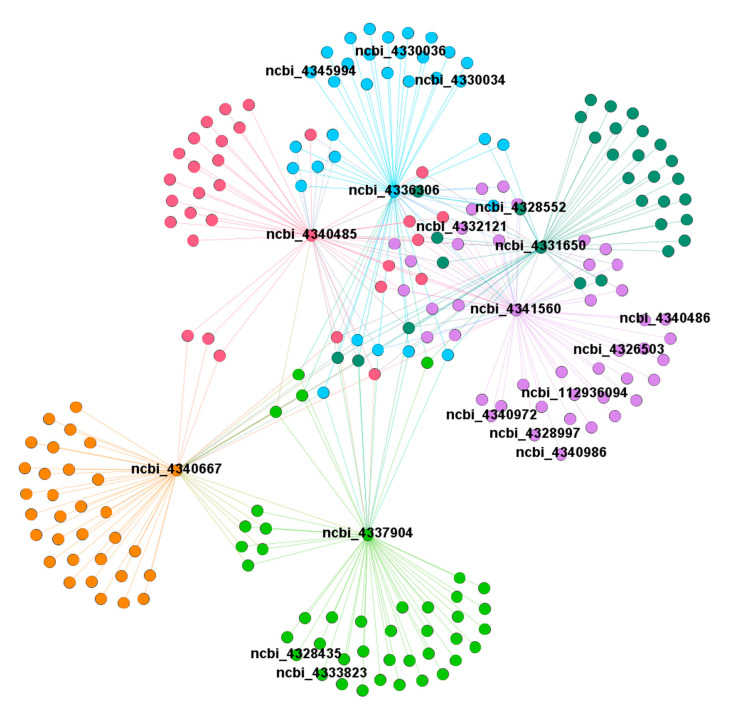
Co-expression regulatory network analysis of module turquoise.

**Figure 12 plants-09-01607-f012:**
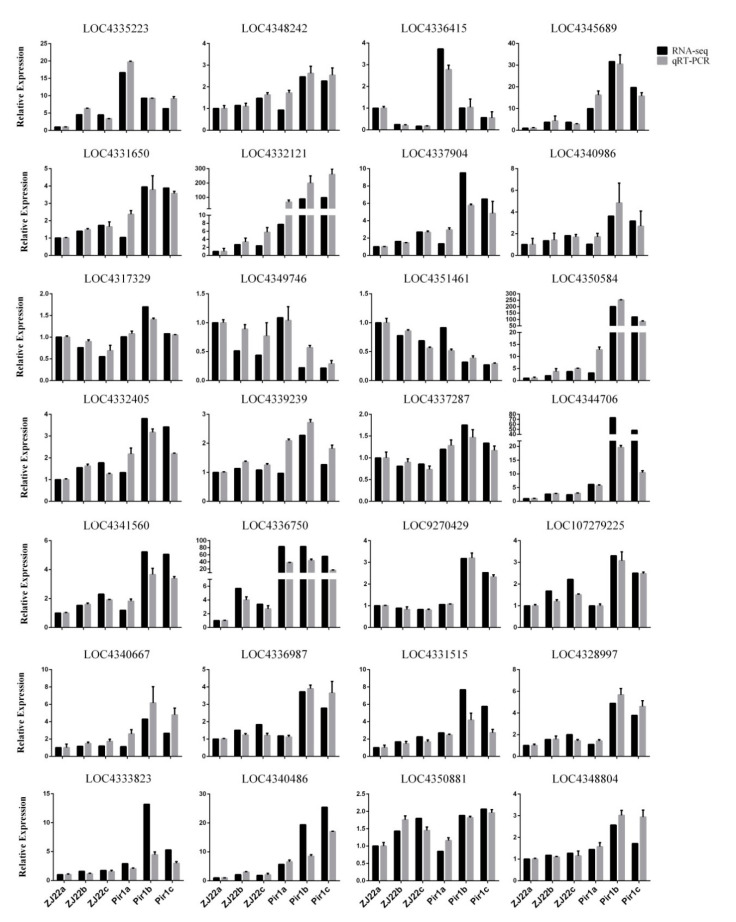
The expression profiles obtained by qRT-PCR and the results of RNA-Seq analysis showing that the two sets of data are well correlated. Results are the means ± standard error of three replications.

**Figure 13 plants-09-01607-f013:**
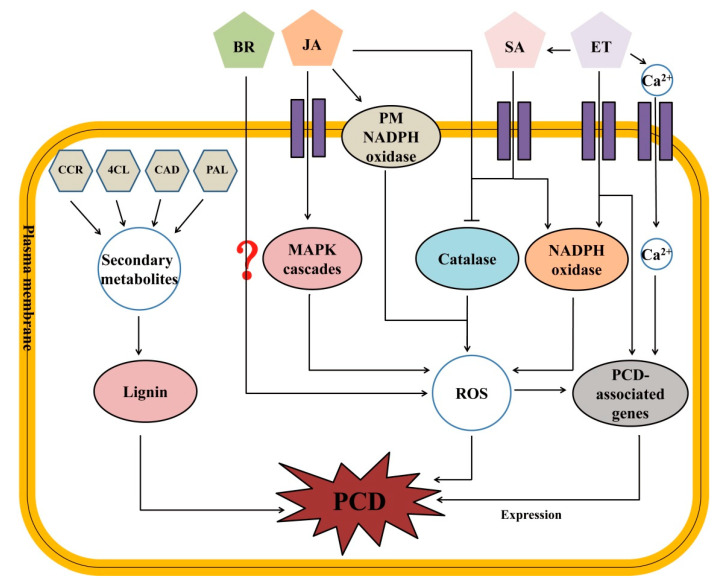
The putative working model for triggering programmed cell death (PCD).

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
