# Peer review of "Gene Mapping, Genome-Wide Transcriptome Analysis, and WGCNA Reveals the Molecular Mechanism for Triggering Programmed Cell Death in Rice Mutant pir1"

_plants, 2020, doi:10.3390/plants9111607_

Round 1
Reviewer 1 Report
The authors have revised the manuscript and the revised version is acceptable
Author Response
Dear Editor-in-chief,
Please see the attachment
Best regards,
Sincerely yours,
Yong Yang

Reviewer 2 Report
Dear Plants Editor in chief,
the reviewer really appreciated authors revisions and accepts the present manuscript after minor revision.
Please consider the following observation:
- The reviewer agrees with the authors that the QTL mapping is unnecessary but strongly recommends to show the obtained maps in Supplementary files.
- Fig. S1 should be moved into the main text.
- Table 1 can be moved into Supplementary materials
- There are some typos in the text. Please check.
Author Response

(The authors gave the same response as above.)

Reviewer 3 Report
I have satisfied with the revised version, with only some minor edits and formats should be made (see in the pdf file). I recommend to publish this paper on the Plants

Author Response
Dear Editor-in-chief,
Please see the attachment
Best regards,
Sincerely yours,
Yong Yang

This manuscript is a resubmission of an earlier submission. The following is a list of the peer review reports and author responses from that submission.
Round 1
Reviewer 1 Report
In this manuscript, the authors carried out phenotypic characterization of lesion mimic mutant followed by RNA-seq analysis. Their manuscript presented some differentially expressed genes and their possible functional characterization. Generally, they provide some information about the mutant in phenotyping and RNA-seq analysis.
However, there is a gap between the observed phenotype and DEGs. There is no solid evidence to show the contribution of any of DEGs to the observed phenotype.
In Figures 1-3, quantitative analysis of phenotype survey should be carried out followed by statistical analysis. For example, in Figure 1, plant height, total biomass, growth period etc should be measured and these data should be subject to statistical analysis to show the difference between control and the mutant.
Mapping analysis showed that a 489kb region with 78 predicted rice genes may be related to the mutant phenotype. DEG analysis should focus on these 78 genes and describe how many of them are differentially expressed and what are their putative functions.
RNA-Seq provide not only info about DEG but also some info about SNP and Indel. The authors should carry out the analysis to figure out these SNPs and Indels between control and the mutant and further investigate the possible contribution to the observed phenotype.
The mutant may show multiple phenotype differences when compared with wildtype, for example, plant height, plant biomass, growth period. The identified DEGs may be related to one or more of these observed phenotypes but the authors only focus on the lesion mimic phenotype.
Author Response

(The authors gave the same response as above.)

Reviewer 2 Report
The authors have made attempts to understand the molecular mechanism involved in PCD. Numerous experiments have been done including leaf histochemical and physiological analyses; mapping the gene locus of the rice mutant; RNA-seq data analysis; co-expression network analysis for module construction; validating the gene expression, etc. The obtained results are of interest to worldwide readers. The current manuscript is well documented and written as a scientific paper. However, I have found some weak points and should be improved to increase the quality of the paper:
- In the introduction part, line 101-102, line 104-108, should be removed to results discussion parts (see in the pdf file)
- Some information should be added the citations (see in the pdf file)
- Some words should be written uniformly (see page 7 and check all manuscript)
- Fig 8 needs to improve high quality and solution
- Page 23, line 271-274 should be removed and added in the discussion part
- Fig 7 page 16, should be added the information of statistical analyses (all figures)
- Check the style and format following the Plants. For instance, the name of genes should be written in italic
- In Materials and Methods: The wild rice type should be added the scientific name; the mutant material should be added the specific information such as line or variety, which offspring of mutation (see in the pdf file)
- Leaf sampling should be added more information (see in the pdf file)
- Page 21 line 448-491, the authors stated that “ a total 736 molecular markers including SSR, STS were screened for polymorphism…….and 236 markers evenly distributed on 12 chromosomes were polymorphic” however, no evidence and information was found, please define or insert directly in line;
- All references need to judiciously check and format following Plants’ style, for example: the name of journals must be made abbreviation etc
- The current paper has a high similarity rate (48%) including references. Authors need to revise and reduce the similarity rate (checked by ithenticate software)

Author Response

(The authors gave the same response as above.)

Reviewer 3 Report
In the manuscript “Gene Mapping, Genome-Wide Transcriptome Analysis and WGCNA Reveals the Molecular Mechanism for Triggering Programmed Cell Death in Rice Mutant pir1”, a comprehensive study to mine the molecular mechanisms triggering programmed cell death (PCD) in rice was conducted coupling gene mapping and a whole transcriptome analyses. A locus controlling PCD, PIR1, was mapped on chromosome 5 and an RNA-Seq was conducted by comparing the transcriptional profiles of leaves of the rice mutant pir1, a spontaneous lesion mimic mutant, and its wild type ZJ22. The analysis revealed the involvement of lignin biosynthesis and plant hormones in the activation of PCD.
The manuscript describes an interesting research that, however, is missing in some important parts and should be extensively revised and improved. The reviewer suggests major revision.
Please consider the following observations:
- As first, the reviewer really appreciated Fig. 17 that summarizes all the obtained results.
- Authors fine mapped the PIR1 locus on rice chromosome 5 using an F2 population including 109 plants and a larger F2 population of 1006 individuals. The obtained map is not reported and the QTL analysis is not described.
- There is not an integration of the two experiments, mapping and RNA-Seq. For example, an analysis of candidate genes at PIR1 locus is missing even though 78 predicted genes are annotated inside the locus on rice genome. A list of these genes is not reported in the text and none of them was considered in the transcriptomic analysis. The reviewer recommends a comparison of the mRNA sequences of each gene in the two genotypes (mutant and wild type). Differences in the sequences could be useful in the developing of molecular markers associated to the genes to be used for co-segregation analyses. Moreover, for each gene, the transcriptional profile in the pair-wise comparisons conducted in the transcriptomic experiment and the presence in the co-expressed regulatory networks must be checked.
- Figures and Tables must be revised and better described in their captions. For example: each genotype must be indicated in each section of Fig. 1; please explain the TB acronym in Fig. 2; please delete the last sentence in Fig. 4 caption; Fig. 5 should be moved to Supplementary files; please explain sample names in Fig. 5 caption; Tables S2-S10 and Tables S11-S20 could be unified, respectively, in single excel files with multiple sheets; Fig. 7 plots should be redone by adding the pval parameters obtained in the GO enrichment analysis; Fig. S4 should be moved to the main text; Figs. 12-15 should be moved to Supplementary files; all Figures and Tables should be moved after the paragraph in which they appear for the first time.
- Please add information about parental lines of the populations indicated in lines 152 and 153.
- In line 170, authors introduced the RNA-Seq experiment. Please add information about the samples utilized for the analysis and about the biological replicates.
- From line 191, authors summarized RNA-Seq results but a clear indication of each sample is missing making the paragraph difficult to read.
- NCBI IDs were utilized for the DEGs classification. The reviewer strictly recommends the use of the nomenclature of the reference genome version utilized in the RNA-Seq experiment.
- A clear indication of the common and specific enriched GO classes in the pair-wise comparisons is missing.
- Which genes have been selected for qRT-PCR validation? Usually validation is performed on the candidate genes discovered.
- Discussion must be extensively revised. It should not represent a further summary of results but a deepened analysis of them on the base of literature.
- To visualized PCD in situ authors used the TB solution. Even if TB is a proper indicator of PCD, a more efficient method is represented by the TUNEL assay. The reviewer suggests to performed also this assay if possible.
- Please explain all the acronyms utilized at the first appearance and in Figures and Tables captions.
- Please indicate the molecular markers database/databases.
- English must be improved. Some sentences are difficult to understand.
Author Response

(The authors gave the same response as above.)
